The efficacy of pre-operative oral aceclofenac and intra-ligamentary mepivacaine on the success of failed inferior alveolar nerve block in patients with symptomatic irreversible pulpitis: a prospective, randomised, double-blinded clinical trial

Ghosh Susmita 1
Gehlot Paras Mull dr.parasmullj@jssuni.edu.in 1
Jadhav Ganesh 2
Mariswamy Annapoorna Ballagere 1
Kishan Karkala Venkappa 3
Al Malwi Ahmed 4
Abullais Shahabe Saquib 5 6
1 Department of Conservative Dentistry and Endodontics, JSS Dental College and Hospital, JSS Academy of Higher Education and Research , Maysore , Karnataka , India
2 Department of Conservative Dentistry and Endodontics, All India Institute of Medical Sciences , Nagpur , Maharashtra , India
3 Srinivas Institute of Dental Sciences, Department of Conservative Dentistry and Endodontics , Mangaluru , Karnataka , India
4 Department of Restorative Dental Science, King Khalid University , Abha , Aseer , Saudi Arabia
5 Department of Periodontics and Community Dental Sciences, College of Dentistry, King Khalid University , Abha , Saudi Arabia
6 Department of Periodontics, Sharad Pawar Dental College, DMIHER , Sawangi , Maharashtra , India
Abu Hasna Amjad
Electronic publication date: 2025 Apr 30
Publication date: 2025
Volume: 13
Electronic Location ID: e19293
Received 2024 Oct 21; Accepted 2025 Mar 19
Copyright: ©2025 Ghosh et al.
Copyright year: 2025
Copyright holder: Ghosh et al.
License: This is an open access article distributed under the terms of the Creative Commons Attribution License, which permits unrestricted use, distribution, reproduction and adaptation in any medium and for any purpose provided that it is properly attributed. For attribution, the original author(s), title, publication source (PeerJ) and either DOI or URL of the article must be cited.
License URL: https://creativecommons.org/licenses/by/4.0/

Keywords: Articaine, Mepivacaine, Intra-ligamentary injection, Aceclofenac, Symptomatic irreversible pulpitis, Failed inferior alveolar nerve block

Funding: The JSS Academy of Higher Education and Research (JSSAHER), Mysuru JSSAHER/REG/URG/54/2023-24/7951/6.10.2023 This research was supported by a Research Grant (sanctioned vide order JSSAHER/REG/ URG/54/2023-24/7951/6.10.2023) from the JSS Academy of Higher Education and Research (JSSAHER), Mysuru. The funders had no role in study design, data collection and analysis, decision to publish, or preparation of the manuscript.

==============================
Objective

This study aimed to compare the effects of oral pre-operative aceclofenac and supplemental intra-ligamentary mepivacaine and articaine injection on the anaesthetic efficacy of failed inferior alveolar nerve block (IANB) in patients with symptomatic irreversible pulpitis in mandibular molars.

Materials and Methods

A total of 126 patients who fulfilled the inclusion criteria were selected for the study intervention. The study design was a prospective, randomised, double-blinded clinical trial. Pre-operative medication was administered 15 minutes before the patients received the standard IANB. Subsequently, the endodontic therapy was started. Pain felt at any time during the procedure was recorded on a 170 mm Heft–Parker visual analogue scale (VAS), and a supplemental intra-ligamentary injection was administered. Based on the pre-operative medication and the intra-ligamentary injection received, the patients were categorized into the following four groups: two groups received 4% articaine with 1:100,000 epinephrine as an intra-ligamentary injection with or without aceclofenac as the pre-operative medication (Group 1 and Group 2, respectively), and two other groups received 3% plain mepivacaine with or without aceclofenac (Group 3 and Group 4, respectively). The VAS scores were determined pre-operatively, following access preparation and/or instrumentation within the dentin, the pulpal space, and the instrumentation of canals. Data were analyzed using Pearson’s chi-square test and independent t-test. A p-value < 0.05 was considered statistically significant.

Results

Concerning the supplemental intra-ligamentary injection, 4% articaine with 1:100,000 epinephrine showed a higher success rate than 3% plain mepivacaine, irrespective of the pre-operative medication used, and the difference was statistically significant (p < 0.05). In addition, if the supplemental intra-ligamentary injection was complemented with pre-operative medication such as aceclofenac 100 mg, the anaesthetic efficacy improved compared to groups receiving a placebo as pre-operative medication (p > 0.05). The anesthetic efficacies of the four groups were as follows: Group 1 (87.1%) > Group 2 (66.7%) > Group 3 (60.6%) > Group 4 (53.1%).

Conclusions

In patients with symptomatic irreversible pulpitis of mandibular molars, an intraligamentary injection of 4% articaine combined with 1:100,000 epinephrine can significantly aid in achieving anesthesia in situations where primary IANB is unsuccessful.

Clinical Relevance

Mandibular molars with symptomatic irreversible pulpitis are associated with a high anaesthetic failure rate of a single primary IANB injection during endodontic treatment. Adding an oral pre-operative aceclofenac medication and a supplemental intra-ligamentary injection, augments the anaesthetic efficacy in such cases.

Introduction

Pain management is a significant concern in dentistry (Raja et al., 2020). Attaining significant pulpal anaesthesia is crucial in endodontic care and dentistry. Anesthesia benefits both patients by providing comfort and dentists by minimizing unexpected reactions or movements during procedures. Achieving anesthesia in mandibular molars with symptomatic irreversible pulpitis may often present challenges (Hargreaves & Keiser, 2002; Claffey et al., 2004). When managing a ‘hot’ tooth, the difficulties escalate. A ‘hot’ tooth, endodontically speaking, is a pulp with an irreversible pulpitis diagnosis, accompanied by spontaneous pain ranging from moderate to severe (Nusstein, Reader & Drum, 2010). The incidence of anesthesia failures associated with inferior alveolar nerve blocks (IANBs) ranges from 44% to 81% (Cohen, Cha & Spångberg, 1993). This high failure rate may be attributed to factors such as accessory innervations, improper injection technique, deflection of the needle, cross innervations and central core theories (Childers et al., 1996). Supplemental anesthesia after initial IANB is often ineffective for endodontic treatment of these specific types of teeth (Gallatin et al., 2000; Nusstein et al., 2003; Moore et al., 2011; Chen et al., 2021). Intra-osseous and intra-ligamentary injections are among the methods that can be employed to precisely deliver a local anaesthetic solution near the root apex (Berlin et al., 2005; Nusstein et al., 2005; Chen et al., 2021).

Administering an anaesthetic solution into the periodontal space via the intra-ligamentary injection technique requires the application of high pressure (Malamed, 1982; Endo, Gabka & Taubenheim, 2008; Moore et al., 2011). Unlike intra-osseous anesthesia, which penetrates the cortical bone, the pressure makes it easier to administer the anesthetic solution into the cancellous bone surrounding the tooth’s root apex (Moore et al., 2011). Other than supplementary injections, previous studies have supported the administration of oral pre-operative medications to enhance the anaesthetic efficacy in patients with symptomatic irreversible pulpitis (Ianiro et al., 2007; Simpson et al., 2011). To manage mild-to-moderate pain stemming from symptomatic irreversible pulpitis, clinicians frequently turn to non-steroidal anti-inflammatory drugs (NSAIDs). These medications block the cyclooxygenase (COX) pathway, halting prostaglandin production. This suppression of prostaglandins is crucial in reducing the inflammation and pain associated with pulpal inflammation (Lamey, 2005). Aceclofenac, chemically known as 2-(2′6′-dichlorophenyl) amino phenylacetoxyacetic acid, is a phenylacetic acid derivative and a potent selective inhibitor of COX-2. Aceclofenac is a well-tolerated NSAID with a lower incidence of gastrointestinal adverse effects than other NSAIDs, such as ibuprofen. This desirable feature improves patient acceptance and compliance (Legrand, 2004). However, limited studies have been conducted on the combined efficacy of a pre-operative medication and a supplemental intra-ligamentary injection. This prospective, randomized, double-blind clinical trial aimed to assess the impact of additional intra-ligamentary injections of articaine or mepivacaine and oral pre-operative aceclofenac on the anesthetic efficacy of failed inferior alveolar nerve blocks (IANB) in patients with symptomatic irreversible pulpitis in lower molars.

Materials & Methods

Study Design: The clinical trial employed a randomized, parallel, double-blind design aimed at demonstrating superiority. The study examined the effects of oral premedication with and without aceclofenac and different intra-ligamentary anesthesia (independent variables) on the success of failed IANB, evaluated using the visual analogue scale (VAS) score (dependent variable). In this setup, both the primary investigator and the patients were kept unaware of the treatment process.

Institutional ethical clearance

The research adhered strictly to the World Medical Association’s Declaration of Helsinki principles. The Institutional Ethics Committee of JSS Dental College and Hospital approved the study’s design and the consent form’s wording in August 2023. (Ethics Committee Registration No. EC/NEW/INST/2022) (JSSDCH IEC Research Protocol No. 35/2022). Written informed consent was obtained from each patient before the commencement of the intervention, which included all the details about the treatment and possible outcomes. To mitigate potential bias in the trial, independent clinicians diagnosed symptomatic irreversible pulpitis.

Protocol registration

This study was registered in the Clinical Trials Registry-India (CTRI) under the identification number CTRI/2023/08/056538 (http://www.ctri.nic.in).

Study setting and location

This study was performed from September 2023 to February 2024 at the Department of Conservative Dentistry and Endodontics (JSS Dental College and Hospital, JSS Academy of Higher Education and Research, Mysore, India).

Sample size calculation

The primary outcome was considered when calculating the sample size. The main goal was to achieve pulp access and canal instrumentation with minimal to no discomfort, which was referred to as success (primary outcome). Failure was, however, characterized as experiencing pain during root canal instrumentation or access preparation, which corresponded to a Heft-Parker VAS score of more than 54 (Heft & Parker, 1984). The sample size was determined using a power analysis program (G*Power, Heinrich Heine University, Düsseldorf, Germany) and data from a previous study, maintaining an alpha level of 0.05 and a beta level of 0.20 for a single-tailed test (Aggarwal et al., 2019). The power analysis indicated that with 30 participants in each group, there would be an 80% chance of identifying a difference in the rate of success by 25%. Although 30 participants were planned for study inclusion, a small number of additional patients were recruited to help compensate for potential dropouts, and maintain adequate power.

Inclusion and exclusion criteria

Inclusion criteria: (i) healthy participants with symptomatic, irreversible pulpitis in mandibular molars. These teeth had to exhibit no periapical changes on intra-oral periapical radiographs (IOPAR) and a pain score >54 on the Heft–Parker visual analog scale (VAS). (ii) Participants had to demonstrate sustained positive responses to thermal and electric pulp tests. (iii) Upon access opening, evidence of vital coronal pulp tissue was required. (iv) According to the American Society of Anesthesiologists, participants’ medical history had to correspond to class I or II. (v) Participants had to be able to comprehend pain scale utilisation.

Exclusion criteria: (i) history of allergy to any component of the anesthetic solution. (ii) Contraindication to any anesthetic solution ingredient. (iii) Patients who are nursing or pregnant. (iv) Current use of pain-relieving medications. (v) Patients experiencing active pain in multiple teeth. (vi) Teeth with fused roots, additional roots, deep periodontal pockets, full crowns, overhanging margins or extensive restorations.

Randomization and blinding method

Randomization: to ensure random assignment, our study employed the dice-roll method to allocate participants in the four study groups.

Blinding Method: in this study, both the patient and the operator were unaware of the treatment assignments. To preserve this blinding, the clinical staff administering the intervention were excluded from the randomization process. An independent team handled the randomization and delivered the intervention in coded packages, including both the pre-operative medication and the intra-ligamentary anesthetic agent. These details were documented in the patient’s history sheet. Pre-operative medications, including a placebo and 100 mg of aceclofenac (Hifenac, Intas Pharmaceuticals Ltd, India), were administered 15 min before IANB. These medications were provided in identical envelopes, each assigned a unique one-digit alphanumeric code generated randomly, using an online tool (https://www.gigacalculator.com/). This code, not derived from a dice-roll method, was solely used to mask the identity of the pre-operative medications.

Cartridges containing the supplemental anaesthetic agent, i.e., 4% articaine with 1:100,000 epinephrine (Septanest, Septodont, France; Batch number: 005821453000) and 3% plain mepivacaine (Scandonest, Septodont, France; Batch number: 005021176000), were covered with a dark tape and labelled with a code different from that assigned for the pre-operative medication. Allocation of patients to groups remained unknown to the operator until the study’s conclusion. Randomization led to the allocation of participants into four different groups:

Group 1 received aceclofenac as pre-operative medication and articaine as intra-ligamentary injection. Group 2 received placebo as pre-operative medication and articaine as intra-ligamentary injection. Group 3 received aceclofenac as pre-operative medication and mepivacaine as intra-ligamentary injection. Group 4 received placebo as pre-operative medication and mepivacaine as intra-ligamentary injection.

Study intervention

A total of 154 patients who visited the Department of Conservative Dentistry and Endodontics at the JSS Dental College and Hospital in Mysore, India, and fit the inclusion criteria participated in the study.

In line with previous studies, before starting treatment, patients used the Heft-Parker VAS to assess their pain level (Claffey et al., 2004). The scale uses a 170-mm VAS line with six categorical labels: faint, weak, mild, moderate, severe, and intense. The extremes of the line are anchored with ’no pain’ on one end and ’unbearable pain’ on the other. Patients received instructions to indicate their pain level by making a mark on the line.

Subsequently, they were asked to take a uniquely coded pre-operative medication 15 min before receiving IANB. Fifteen minutes later, standard inferior alveolar nerve block anesthesia was administered to the patients. This involved injecting 1.8 mL of a 2% lidocaine solution containing 1:80,000 epinephrine (Lignox 2% A, Indoco Remedies Ltd, Mumbai, India) via the direct Halsted approach. After drying the injection site with sterile gauze, a 60-second application of 2% lidocaine gel (LOX-2% Jelly, Neon Laboratories, Mumbai, India) was applied topically using a sterile cotton swab. Next, the solution was injected using a 2.5-mL disposable syringe with a 38-mm 26-G needle (Unolok, Hindustan Syringes and Medical Devices LTD, Faridabad, India). The clinician carefully aligned the needle with an imaginary line extending from the coronoid notch to the pterygomandibular raphe, positioning the tip two mm above the occlusal plane of the mandibular molars. The syringe barrel was stabilized between the molars on the opposite side of the mouth from where the injection was given. The needle was carefully inserted until it hit the bone. Once the correct location was reached, fluid was drawn back through the needle to ensure it was in the right place, and then the solution was slowly injected over 2 min. Patients were examined for lip numbness ten minutes following the injection. If a patient did not have complete numbness, the block was considered unsuccessful, and the patient was not included in the study.

After excluding six patients without profound lip numbness, the study included 148 patients with this condition. Each underwent a conventional access opening under rubber dam isolation on the affected tooth. If any pain occurred during the procedure, the treatment was stopped, and the pain was assessed using the Heft–Parker VAS.

The extent of access preparation and/or instrumentation was documented as within the dentin, the pulpal space, and the instrumentation of canals. During their treatment, 126 of the 148 initial patients had moderate to severe pain (defined as Heft-Parker VAS ratings of more than 54). For patients whose initial inferior alveolar nerve block (IANB) was ineffective, additional intra-ligament injections were administered as an intervention. These injections were prepared by an independent team, who concealed the cartridge with dark tape and coded it with a unique numeric code as described earlier.

The rubber dam was removed. A pressure syringe (MediJect, Medidenta, Las Vegas) fitted with a 30-gauge, 12-mm short needle slightly curved for ease of insertion was used to administer intraligamentary injections. At the tooth’s mesiobuccal line angle, between the alveolar bone and the tooth, the needle was firmly inserted into the mesial gingival sulcus until resistance was felt. The needle was angled thirty degrees concerning the long axis of the tooth. One dose (0.2 mL) was delivered by applying intense back pressure while the handle/trigger was entirely squeezed. Injections were repeated until back pressure was felt, repositioning the needle if necessary.

A 0.6 mL of anaesthetic solution (three doses from the pressure syringe) was given. Following the injection, the pressure and needle were left in place for twenty seconds. The same procedures were carried out for the distal root with an injection made at the distobuccal line angle into the gingival sulcus.

Following the deposit of 1.2 mL of the solution, the needle was carefully loosened and withdrawn from the gingival sulcus. Reapplying the rubber dam permitted the endodontic treatment to continue. Success was measured by the patient experiencing little to no pain (Heft-Parker VAS score ≤ 54 mm) during the access preparation and instrumentation stages. The Consolidated Standards of Reporting Trials (CONSORT) flowchart diagram of randomized controlled protocol is shown in Fig. 1.

Figure 1 CONSORT flow diagram.

Data evaluation and statistical analysis

Data were tabulated and statistically analysed using the SPSS 23.0. (Statistical Package for the Social Sciences, ver 23.0, IBM Corp, United States). The participants’ ages and initial mean VAS scores were outlined by calculating the means and standard deviations, and these were computed using repeated measures analysis of variance (ANOVA). Pearson’s chi-square test compared gender, tooth type, and anaesthetic success across the four groups. Statistical significance was set at p < 0.05. A t-test compared the success rates among the four groups.

Results

The CONSORT diagram (Fig. 1) shows that 154 mandibular molars with symptomatic irreversible pulpitis were assessed for eligibility, of which six showed inadequate lip numbness and were considered as ‘missed’. Thus, 148 teeth remained, of which 22 IANB injections were successful (15% success rate), which resulted in 126 teeth with failed IANB being randomized for allocation into four groups. The demographic data (age, gender, tooth type, and initial VAS score) of the study population in each group are summarized in Table 1. There were no statistically significant differences among the groups regarding the demographics; therefore, the equality of the groups was justified.

Table 1 Demographic and clinical characteristics between the groups and initial VAS.

Characteristics		Group 1	Group 2	Group 3	Group 4	p-value	
Age	Range	19–51	19–63	18–61	20–64	0.33*	
Mean ± SD	33 ± 9.21	37 ± 12.63	37 ± 13.0	37 ± 12.5	
Gender	Male	15	14	14	15	0.97**	
Female	16	16	19	17	
Molar type	First	16	16	16	17	0.98**	
Second	15	14	17	15	
Initial VAS	Mean ± SD	123.66 ± 11.54	124.38 ± 11.78	125.18 ± 12.2	122.19 ± 12.44	0.95*	
Notes.

SD Standard Deviation

VAS Visual Analog Scale

P-value, significant (p ≤ 0.05).

* ANOVA F = 1.158, ANOVA F = 0.125 (Age).

** Pearson χ2 = 0.25 (Gender), Pearson χ2 = 0.194 (Molar type).

Table 2 presents the success and failure rates for the four groups. Group 1, when compared to Group 2 and Group 3, had a statistically significant difference (p = 0.048 and p = 0.02 respectively).

Table 2 The success and failure rate (%) in various groups.

	Group 1
n (%)	Group 2
n (%)	Group 3
n (%)	Group 4
n (%)	Total
n (%)	
Success	27 (87.1)	20 (66.7)	20 (60.6)	17 (53.1)	84 (66.7)	
Failure	4 (12.9)	10 (33.3)	13 (39.4)	15 (46.9)	42 (33.3)	
Total	31 (100)	30 (100)	33 (100)	32 (100)	126 (100)	
Notes.

n number of subjects

Pearson χ2 = 9.009, p = 0.03.

Figure 2 depicts the success rate of articaine intra-ligamentary injections with and without aceclofenac as 44.27% and 32.73%, respectively, and for mepivacaine intra-ligamentary injections with and without aceclofenac as 30.78% and 26.12%, respectively. Independent samples t-tests were performed to statistically compare anesthetic success rates between the treatment groups, as shown in Table 3.

Figure 2 The success rate of the intra-ligamentary injections of articaine and mepivacaine with aceclofenac and placebo.

Table 3 Pairwise comparison of anaesthetic success between the treatment groups.

Groups	n	Mean VAS (± SD)	p-value	
Group 1
Group 2	31
30	18.65 ± 12.28
30.03 ± 13.63	<0.05*	
Group 3
Group 4	33
32	31.68 ± 13.36
34.79 ± 14.27	>0.05	
Group 1
Group 3	31
33	18.65 ± 12.28
31.68 ± 13.36	<0.05*	
Group 2
Group 4	30
32	30.03 ± 13.63
34.79 ± 14.27	>0.05	
Notes.

n number of subjects

SD Standard Deviation

* Statistically significant difference at p-value < 0.05 (Independent t-test).

Discussion

In this study, primary IANB exhibited a high failure rate of 85%. Previous studies by Aggarwal et al. have also reported high failure rates of 70%–80% (Aggarwal, Jain & Kabi, 2009; Aggarwal et al., 2015). Lapidus et al. (2016) explored the potential benefits of pre-operative medication as an addition to IANB for patients with irreversible pulpitis. Their research indicated moderate evidence that taking oral non-steroidal anti-inflammatory drugs (NSAIDs) before receiving IANB local anaesthetic (1.8–3.6 mL of 2% lidocaine) could offer additional pain relief (Lapidus et al., 2016). Studies have examined the effects of pre-treatment administration of aceclofenac, ibuprofen, or paracetamol on the effectiveness of local anesthetic in patients with irreversible pulpitis. The findings revealed a success rate of 93% with ibuprofen, 90% with aceclofenac and 73% with paracetamol (Ramachandran et al., 2012). In this study, the group receiving articaine showed a better success rate regarding the analgesic effect when administered with aceclofenac (i.e., Group 1 with 87% success) than that receiving the placebo (i.e., Group 2 with 66% success). The success rates of the two groups differed significantly (p = 0.048). Furthermore, similar results were noted in the two mepivacaine groups, wherein Group 3 receiving aceclofenac showed 60% success, and Group 4 receiving the placebo exhibited 53% success. However, the difference was not statistically significant (p = 0.595). Apart from pre-operative medications, the utilization of intra-ligamentary injection as a supplementary method has been validated by several studies. A survey conducted among endodontists in the United States revealed that most clinicians preferred using intra-ligamentary injection as the supplemental injection (Bangerter, Mines & Sweet, 2009). In a study by Walton and Abbott, the efficacy of intra-ligamentary injections was evaluated in 120 patients who lacked sufficient pulpal anaesthesia. Anesthesia took effect almost immediately, with an overall success rate of 92% (Walton & Abbott, 1981). Intra-ligamentary injections resulted in a 70% success rate for patients with unsuccessful IANBs, according to a study by Zarei et al. (2012). Aggarwal et al. (2019) conducted a study comparing 4% articaine and 2% lidocaine as additional injections after an IANB did not work. The articaine injections were successful for 66% of patients, while the lidocaine injections worked for 78% of patients. Aggarwal et al. (2024) conducted a study examining the use of tramadol, both on its own and combined with 2% lidocaine, as an additional intra-ligamentary injection to support anesthesia. The results suggested that supplementing 2% lidocaine with 1:80,000 epinephrine by adding tramadol as an intra-ligamentary injection can improve anesthesia success during endodontic treatment of mandibular molars with irreversible pulpitis that are resistant to IANB. In clinical research, diverse methods are employed to evaluate the efficacy of pulpal anaesthetics, including assessing numbness in soft tissues, lips and tongue tips. Nevertheless, interpreting pain experienced by the patients adds complexity to the study as it is highly subjective and is influenced by emotional and psychological factors and past experiences, complicating the evaluations (Certosimo & Archer, 1996). After each injection, all participants in this study experienced lip numbness. All patients experienced lip numbness, but everyone did not obtain pulpal anaesthesia. Hence, lip numbness was considered a definitive sign to ensure the administration of an accurate block and further continue the study. Moreover, the VAS, a tool for gauging subjective experiences such as dental pain, was utilized as a measurement device. This scale has proven effective in dental practice, especially for patients experiencing symptoms before treatment (Heft & Parker, 1984; Aggarwal, Jain & Kabi, 2009; Aggarwal et al., 2015; Aggarwal et al., 2019). In this study, the overall success rate of the articaine group (Group 1 and Group 2) was 77%, which was better than that of the mepivacaine group (Group 3 and Group 4), with a success rate of 56.9%, irrespective of the pre-operative medication administered, and the difference was statistically significant (p < 0.05). It was also observed that the intra-ligamentary injections work better when supplemented with a pre-operative medication. This study found that an additional articaine intra-ligamentary injection and aceclofenac provide more predictable anaesthesia than the conventional IANB used alone in managing patients with symptomatic irreversible pulpitis. Nevertheless, apprehension, anxiety and stress commonly experienced by patients with dental pain can significantly influence the results. One potential limitation of this study is the variation in epinephrine concentrations between the solutions. Epinephrine has been demonstrated to influence anaesthetic efficacy, so using plain mepivacaine could have introduced bias in the study (Aggarwal et al., 2019). Moreover, the sample size was determined as per previous studies; however, a larger sample size would have a greater impact in validating the study results. Another limitation of this study is that post-operative pain levels at 6, 12 and 24 h were not recorded. These data could have explained how effectively the pre-operative medication and the anesthetic solution worked post treatment.

Conclusions

Within the limitations of this in vivo investigation, it can be concluded:

Combining a pre-operative oral medication of aceclofenac and a supplemental intra-ligamentary injection of 4% articaine showed superior results, compared to 3% plain mepivacaine. However, further studies must be conducted with other types of supplemental anaesthetic techniques so that the patient can experience a pain-free endodontic treatment.

Supplemental Information

Supplemental Information 1 Master sheet

Supplemental Information 2 CONSORT checklist

The authors thank the clinical staff and research team for their support in conducting the study.

Additional Information and Declarations

Competing Interests

Author Contributions

Human Ethics

Clinical Trial Ethics

Data Availability

Clinical Trial Registration

The authors declare there are no competing interests.

Susmita Ghosh conceived and designed the experiments, performed the experiments, prepared figures and/or tables, authored or reviewed drafts of the article, and approved the final draft.

Paras Mull Gehlot analyzed the data, prepared figures and/or tables, and approved the final draft.

Ganesh Jadhav conceived and designed the experiments, performed the experiments, analyzed the data, authored or reviewed drafts of the article, and approved the final draft.

Annapoorna Ballagere Mariswamy conceived and designed the experiments, performed the experiments, prepared figures and/or tables, authored or reviewed drafts of the article, and approved the final draft.

Karkala Venkappa Kishan performed the experiments, analyzed the data, prepared figures and/or tables, and approved the final draft.

Ahmed Abdullah Al-Malwi performed the experiments, analyzed the data, authored or reviewed drafts of the article, and approved the final draft.

Shahabe Saquib Abullais conceived and designed the experiments, prepared figures and/or tables, and approved the final draft.

The following information was supplied relating to ethical approvals (i.e., approving body and any reference numbers):

The Institutional Ethics Committee of JSS Dental College and Hospital approved the study’s design and the consent form’s wording in August 2023. (Ethics Committee Registration No. EC/NEW/INST/2022) (JSSDCH IEC Research Protocol No. 35/2022).

The following information was supplied relating to ethical approvals (i.e., approving body and any reference numbers):

Institutional Ethics Committee of JSS Dental College and Hospital.

The following information was supplied regarding data availability:

The raw data is available in the Supplemental File.

The following information was supplied regarding Clinical Trial registration:

CTRI/2023/08/056538.

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
