# Peer review of "The efficacy of pre-operative oral aceclofenac and intra-ligamentary mepivacaine on the success of failed inferior alveolar nerve block in patients with symptomatic irreversible pulpitis: a prospective, randomised, double-blinded clinical trial"

_PeerJ, doi:10.7717/peerj.19293_

## Round 0.1 · original submission · Major Revisions

Dear authors,

Your manuscript was reviewed by 4 external reviewers. The manuscript requires substantial revision to address critical issues in methodology, study design, and reporting. Key concerns include unclear definitions of primary and secondary outcomes, inadequate sample size calculation aligned with study objectives, and a confusing description of the randomization and blinding process. The Materials and Methods section must clearly define groups, interventions, and success criteria while adhering to CONSORT guidelines. Results should be more descriptive, exploring statistical analyses, group demographics, and intervention characteristics. Figures and tables need enhanced clarity and alignment with text. Addressing these issues and providing robust comparisons with similar studies will strengthen the manuscript's validity and clinical relevance.

Reviewer 1 ·

Basic reporting

Important issues were observed in the methodology of the study. The authors should carefully revise the study report.

Material and Methods:
- Line 132-133: The authors should add an item entitled “study design” to describe the study characteristics as well as determine the independent and dependent variables.
- Line 133: What do the authors mean by “both primary investigator and the patient remained unaware of the process”? Is it related to the blinding? If so, please describe this information in a separate item “blinding”.
- Sample size calculation: There is a misunderstanding about the concept of determining the sample size using the primary and secondary outcomes. Firstly, the authors did not mention the primary and secondary outcomes. The sample size calculation is based only on the primary outcome of study. If so, how was the parameters filled out on the G*Power? It is not clear the if the “data from a previous study” is a primary or secondary outcome. Which were data used? Additionally, the authors classified the study as a superiority clinical trial, and then, the sample size calculation needs to follow this design, which was not followed by calculating it on the G* Power.
- Randomization method: add the information of blinding in a separate item. The randomization process by the dice-roll method is not properly described. If the clinical staff used the method, how was the intervention implemented? Afterwards, the authors mentioned that the “placebo” or “aceclofenac” was randomly identified as a unique one-digit alphameric code. Is this code provided by the dice-roll method? It is not clear, and the randomization process is confusing.
- Lines 185-188: this information should be described in the “institutional ethical clearance”.
- In the “randomization method”, the authors mentioned that supplemental anesthetic agent such as 4% articaine and 3% mepivacaine were administered, but the intervention was described using 2% lidocaine. So, in every case, was the lidocaine administered? Was supplemental anesthesia required for all cases? It is not clear what was conducted. The authors should carefully revise the study design, the purpose, and the intervention. Additionally, the authors mentioned that the CONSORT flowchart shows the participant enrollment, but is this guideline used to report the study? I strongly recommend that the authors revise the study report following the CONSORT. The guidelines also recommends that the flowchart must be reported in the “results” section.

Results:
- The results should be more descriptive. The authors should explore the results.
- What was considered success in the study? This should be determined in “material and methods”. How was it calculated?
- In the figure 2 is now clear what was performed in the study. Then, I recommended that the authors revise the text and explicit clearly this information.
- The authors determined in table 2 the four groups, but there is no description of them before.

Experimental design

Please see my comments in "basic report".

Validity of the findings

Please see my comments in "basic report".

Reviewer 2 ·

Basic reporting

1. Clear and professional language used
2. Good structure and thorough analysis of the intervention
3. Sample size calculation and ethical approval mentioned
4. Results section requires further improvement : Clarity over the characteristics of intervention in each group. Comment on the equality of the groups in regards to demographics as shown in Figure 1 has to be added in the text. Further description on the groups that have been compared has to be added in the results section. The tables without any statistical tests are weak
5. Figure 2 may require better analysis for printing

Experimental design

1. Aim of the study well presented
2. Protocol description is good
3. Groups need to be defined more clearly in the method section

Validity of the findings

Good overall discussion

·

Basic reporting

No comment

Experimental design

Clarification is needed about sample size.
It has been mentioned that 30 participants will be allocated to each group, but extra participants were added afterwards. Additionally , why the drop out rate was not calculated?, also please mention the effect size.

Validity of the findings

No comment based on justification of the sample size selected.

Additional comments

No comments.

Reviewer 4 ·

Basic reporting

This is written in the PDF

Experimental design

This is written in the PDF.

Validity of the findings

This is found in the PDF.

Additional comments

-Emphasizing the clinical relevance of the findings would strengthen the manuscript.
-additional details on the randomization process and blinding protocol, particularly regarding the preparation and administration of anesthetic agents, would enhance reproducibility and transparency.
-While the discussion references previous studies, including more comparisons with similar research on aceclofenac or alternative NSAIDs would provide a broader context for interpreting the findings
-other comments can be found in the PDF attached.

Annotated reviews are not available for download in order to protect the identity of reviewers who chose to remain anonymous.

---

## Round 0.2 · accepted · Accept

Dear authors,

We appreciate your thorough revisions and for addressing all the reviewers' comments. We are pleased to inform you that your manuscript has been accepted for publication.

Congratulations on this achievement! We appreciate your contribution and look forward to sharing your work with the scientific community.

Reviewer 4 ·

Basic reporting

I am satisfied with the modifications.

Experimental design

I am satisfied with the modifications.

Validity of the findings

I am satisfied with the modifications.

Additional comments

I am satisfied with the modifications.